# Gestational Diabetes-like Fuels Impair Mitochondrial Function and Long-Chain Fatty Acid Uptake in Human Trophoblasts

**DOI:** 10.3390/ijms252111534

**Published:** 2024-10-27

**Authors:** Kyle M. Siemers, Lisa A. Joss-Moore, Michelle L. Baack

**Affiliations:** 1Sanford School of Medicine, University of South Dakota, 414 E. Clark Street, Vermillion, SD 57069, USA; kyle.siemers@coyotes.usd.edu; 2Department of Pediatrics, University of Utah, 295 Chipeta Way, 2N131, Salt Lake City, UT 84108, USA; lisa.joss-moore@hsc.utah.edu; 3Department of Pediatrics, Division of Neonatology, Sanford School of Medicine, University of South Dakota, 1400 W. 22nd St., Sioux Falls, SD 57105, USA; 4Environmental Influences on Health and Disease Group, Sanford Research, 2301 E. 60th St., Sioux Falls, SD 57104, USA

**Keywords:** fatty acids, trophoblast, mitochondria, lipid droplets, gestational diabetes mellitus

## Abstract

In the parent, gestational diabetes mellitus (GDM) causes both hyperglycemia and hyperlipidemia. Despite excess lipid availability, infants exposed to GDM are at risk for essential long-chain polyunsaturated fatty acid (LCPUFA) deficiency. Isotope studies have confirmed less LCPUFA transfer from the parent to the fetus, but how diabetic fuels impact placental fatty acid (FA) uptake and lipid droplet partitioning is not well-understood. We evaluated the effects of high glucose conditions, high lipid conditions, and their combination on trophoblast growth, viability, mitochondrial bioenergetics, BODIPY-labeled fatty acid (FA) uptake, and lipid droplet dynamics. The addition of four carbons or one double bond to FA acyl chains dramatically affected the uptake in both BeWo and primary isolated cytotrophoblasts (CTBs). The uptake was further impacted by media exposure. The combination-exposed trophoblasts had more mitochondrial protein (*p* = 0.01), but impaired maximal and spare respiratory capacities (*p* < 0.001 and *p* < 0.0001), as well as lower viability (*p* = 0.004), due to apoptosis. The combination-exposed trophoblasts had unimpaired uptake of BODIPY C12 but had significantly less whole-cell and lipid droplet uptake of BODIPY C16, with an altered lipid droplet count, area, and subcellular localization, whereas these differences were not seen with individual high glucose or lipid exposure. These findings bring us closer to understanding how GDM perturbs active FA transport to increase the risk of adverse outcomes from placental and neonatal lipid accumulation alongside LCPUFA deficiency.

## 1. Introduction

Gestational diabetes mellitus (GDM) is caused by excessive insulin resistance during pregnancy, resulting in increased levels of circulating glucose and lipids in the parent [1]. GDM affects nearly 14% of pregnancies worldwide and its prevalence is increasing [1,2,3,4]. The current standard of care focuses on treating the resulting hyperglycemia during GDM, which reduces the perinatal mortality [5], but does not reduce the risk of all adverse outcomes. Notably, macrosomia and long-chain polyunsaturated fatty acid (LCPUFA) deficiency in offspring can persist despite good blood glucose control [6,7]. In the short term, macrosomia can cause birth trauma and increased rates of preterm birth and C-sections [2,8,9,10]. In the long term, macrosomia and LCPUFA deficiency are associated with a higher lifetime risk of neurodevelopmental disorders, metabolic syndrome, insulin resistance, diabetes, obesity, and cardiovascular disease [7,11,12,13,14]. The persistence of these adverse outcomes despite good glycemic control suggests that additional circulating fuels such as lipids may play a role.

Pregnancy is a progressively anabolic state associated with decreasing insulin sensitivity and increasing hyperlipidemia that culminates in the third trimester to meet the high metabolic needs of the parent, the placenta, and the rapidly growing fetus [15,16]. This physiologic hyperlipidemia of pregnancy is crucial for the near-term fetus. Specifically, essential LCPUFAs are needed to support rapid growth and brain development and cannot be made de novo by the fetus. Because the fetus relies on a parental source of LCPUFAs, the placenta plays a distinct and crucial role in their transport. Indeed, the placenta mediates the biomagnification of LCPFUAs so that the third-trimester levels can be higher in the fetus than parent [17]. The placenta’s biomagnification of LCPUFA not only highlights the developmental importance of these essential fatty acids (FA), but also the need for energetically demanding active transport [18].

Mitochondria are critical regulators of active lipid transport, as well as many other functions in the placenta. The mitochondria are not only the primary ATP producer, but also the primary metabolizers of FA. Mitochondria are highly dynamic and can undergo replication and biogenesis in response to metabolic cues. By these mechanisms, mitochondria rapidly increase or decrease in number and function to meet energetic needs and maintain cellular homeostasis under changing metabolic conditions such as nutrient availability. In turn, FAs regulate mitochondrial biogenesis in a feedback loop that is sensitive to lipid availability, including when they are in excess, as in pregnancy [19,20]. When the balance between FA availability and mitochondrial function is tipped, lipid accumulates in droplets and organelles [21,22]. This lipotoxicity can drive further mitochondrial dysfunction and reactive oxygen species (ROS) production to influence cell fate by signaling cell-cycle progression, apoptosis, and necrosis [23], which can increase inflammation to impair angiogenesis [24] and further impact placental structure and function.

How the placenta processes and transports lipids is of particular interest in the context of GDM, a state of exaggerated nutrient excess. As early as the first trimester, parents who go on to be diagnosed with GDM have higher levels of circulating lipids than those who do not [25,26]. By the third trimester, the parent’s lipid levels are more predictive of infant macrosomia and adiposity than glucose, parental weight gain, or pre-pregnancy body mass index (BMI) [27]. Despite higher lipid stores, infants exposed to GDM have a relative deficiency of essential LCPUFAs [28,29,30]. Kinetic studies utilizing stable isotopes to trace FA transport from parents to their fetuses highlight the fact that multiple lipid species in parental circulation contribute to FA transfer, and that GDM selectively disrupts the transfer of docosahexaenoic acid (DHA), an essential LCPUFA, while other FAs are accreted [31,32,33]. Interestingly, placentas in patients with GDM also accumulate more triglyceride species but are notably deficient in LCPUFAs [33]. This supports the prevailing theory that LCPUFA uptake into the placenta is the catch point that causes a selective LCPUFA deficiency in the fetus [34].

For these reasons, it is important to study the effects of diabetic fuels on the trophoblast metabolism and the kinetics of various FA species. Previous studies have examined the separate effects of glucose or individual FA species such as lipotoxic palmitate on trophoblast metabolism and cell fate [35,36,37,38], but GDM increases the parental levels of many FAs, and individual FAs have distinct biological features, depending on their acyl chain length (number of carbons) and saturation (number of double bonds), that could impact their transport into cells [39]. To our knowledge, there have been no studies that examine the effects of both glucose and lipid overload on trophoblast metabolism and species-specific FA uptake.

The objective of this study is to determine the impact of high glucose conditions, high lipid conditions, and their combination on trophoblast metabolism and species-specific FA uptake. We hypothesize that the combination of excess glucose and lipids impairs mitochondrial function and inhibits the uptake of FAs with longer acyl chains in human trophoblasts. To test our hypothesis, we used both BeWo, a choriocarcinoma cell line, and primary isolated cytotrophoblasts (CTBs). The strengths of the BeWo model include an ability to isolate the effects of fuel exposures without introducing confounding variables associated with individual pregnancies, as well as the ability to use longer experimental timelines compared to CTBs, which spontaneously differentiate and syncytialize into less metabolically active syncytiotrophoblasts (SCT) after 72 h [40]. While others have compared facets of the form and function of BeWo and CTBs [41], we included both of these cell types to present a comparison of their baseline uptake of FA and to determine the suitability of using BeWo to measure effect of diabetic fuels. To understand the responses of BeWo to pooled lipid overload, our high lipid conditions entailed the use of a lipid emulsion with a defined mixture of cholesterol and various species of FA, including myristic (14:0), palmitic (16:0), palmitoleic (16:1), stearic (18:0), oleic (18:1), linoleic (18:2), linolenic (18:3), and arachidonic (20:4) FAs, rather than testing the effects of individual FAs that can be lipotoxic. We also examined the effects of combined high glucose and high lipid exposure, hereafter called diabetic fuels, on the BeWo uptake of FA species that vary by carbon length and saturation, then characterized their accumulation in lipid droplets and assessed the lipid droplet dynamics. Overall, this study leveraged BeWo to show that the combination of high glucose and high lipid conditions impairs mitochondrial function and the uptake of the longest FA species that we tested, and also showed that the species-specific FA uptake is similar in primary CTB from term pregnancies.

## 2. Results

### 2.1. The Combination of High Glucose and High Lipid Exposure Impairs BeWo Viability

Based on 4-day growth curves, there were no fuel-mediated differences in BeWo growth (Figure 1A), doubling time, or fold change (Figure 1B). However, using flow cytometry to quantify the cells that stained positive for Annexin V (APC-A), an apoptosis marker, and propidium iodide (PE-A), a marker for cell death (Figure 1C), we showed that BeWo exposed to the combination of high glucose plus high lipid conditions had significantly fewer viable cells compared to controls (*p* = 0.004, *n* = 3/group) and compared those exposed to high glucose (*p* = 0.012) or high lipid (*p* = 0.005) alone. The combination-exposed BeWo also had more apoptotic cells than controls (*p* = 0.0258) and the high lipid-exposed BeWo (*p* = 0.027) (Figure 1D). Overall, the diabetic fuels did not signficiantly impact growth, but instead impacted BeWo viability.

### 2.2. High Lipid Exposure Increases Mitochondria Abundance in BeWo

To examine the effect of diabetic fuels on mitochondria abundance, we used western blot to probe for the mitochondrial proteins translocase of the outer mitochondrial membrane 20 (TOM20) and voltage-dependent anion channel (VDAC) in protein lysate from BeWo cells exposed to control, high glucose, high lipid, and combination media for 72 h (Figure 2A). Both the high lipid and combination conditions increased the abundance of TOM20 relative to β-actin (*p* = 0.0457 and 0.0099, respectively) (Figure 2B). The combination exposure also increased the VDAC abundance (*p* = 0.0236) (Figure 2C). The ratio of VDAC to TOM20 was preserved (Figure 2D), suggesting no change in the VDAC abundance when normalized to the amount of mitochondria as measured by TOM20. Taken together, the high lipid conditions increased the abundance of mitochondria in BeWo.

### 2.3. The Combination of High Glucose and High Lipid Exposure Impairs Mitochondrial Respiration in BeWo

Seahorse extracellular flux (XF) analysis was used to evaluate the effects of diabetic fuels on BeWo bioenergetics. A plot of the oxygen consumption rate (OCR) over the course of a mitochondrial stress test demonstrated very little effect of high glucose or high lipid conditions alone, but that the combination markedly reduced mitochondrial respiration in the BeWo cells (Figure 3A). When quantified, the basal respiration was not statistically different (Figure 3B), but the cells exposed to the combination of high glucose and high lipid conditions had maximal respiration approximately half that of controls (*p* = 0.0007), and respiration that was lower than those exposed to high glucose (*p* = 0.0489) or high lipid conditions alone (*p* = 0.0102) (Figure 3C). Further, the combination exposure depleted the spare respiratory capacity by 91% compared to controls (*p* < 0.0001) (Figure 3D) and by over 85% compared to the individual exposures (*p* = 0.0011 vs. high glucose, *p* = 0.0007 vs. high lipid). There were no differences in glycolysis (Figure 3E), maximum glycolysis (Figure 3F), or spare glycolytic capacity (Figure 3G) when compared to controls, but the spare glycolytic capacity was lower in the combination-exposed BeWo cells compared to those exposed to high glucose alone (*p* = 0.0240). The calculated ATP production was not different between groups (Figure 3H). The proton efflux rate (PER) was calculated to estimate the effects of diabetic fuels on anaerobic (contribution from lactate) (Figure 3I) and aerobic (contribution from CO_2_) (Figure 3J) during glycolysis. The combination-exposed BeWo had a significantly lower PER from aerobic glycolysis compared to controls (*p* = 0.0077), as well as those exposed to high glucose or high lipid conditions alone (*p* = 0.0421 and *p* = 0.0126, respectively). In summary, while individual fuels had little effect on BeWo cells’ bioenergetics, the combined exposure significantly impaired mitochondrial respiration without increasing the glycolytic reserve, resulting in little to no spare capacity.

### 2.4. Fatty Acid Uptake by BeWo Is Carbon Length and Saturation Dependent

Using 4,4-difluoro-3a,4a-diaza-s-indacene (BODIPY)-labeled FA analogues, BODIPY C12, C16, and monounsaturated (MU) C12 and confocal imaging at 5, 20, 60, 120, and 180 min we determined the rates of FA uptake in control BeWo cells over time (Figure 4A). The length of the BODIPY fluorophore probe is constant and approximates the length of a four-carbon chain [42]. BODIPY can be metabolized into varying polar and non-polar lipids [42]. Thus, BODIPY C12 approximates a saturated 16-carbon FA, BODIPY C16 approximates a saturated 20-carbon FA, and BODIPY MU C12 approximates a 16-carbon FA with one double bond in the acyl chain. The fluorescent intensity within whole cells was quantified using Image J (Version 1.54f), averaged across replicates, and plotted to highlight the variability in FA uptake kinetics related to four additional carbons or one double bond (Figure 4B). Specifically, BODIPY C12 was taken up rapidly and peaked at 20 min, whereas BODIPY C16 was taken up more slowly with a peak at 180 min. BODIPY MU C12, which has an additional double bond compared to C12, had significantly less whole-cell uptake across the experiments, with fluorescence between 0.03% (at 60 min) and 3% (at 5 min) copared to that of its saturated counterpart. The kinetics of BODIPY MU C12 also varied, with rapid and late biphasic peaks around a nadir at 60 min. Taken together, the rates and patterns of FA uptake in BeWo vary based on the acyl chain length and saturation. Longer-chain FAs are taken up more slowly and a double bond markedly diminishes uptake.

### 2.5. Excess Glucose and Lipids Impact Whole-Cell BODIPY C16 and MU C12 Fatty Acid Uptake Without Impairing BODIPY C12 Uptake

The uptake of BODIPY C12 was not changed by culturing the BeWo cells in high glucose, high lipid, or combination media (Figure 5A). However, diabetic fuels altered the whole-cell uptake of BODIPY C16 (Figure 5B) and BODIPY MU C12 (Figure 5C). Specifically, there was ~14% less BODIPY C16 accretion at 180 min with either high glucose (*p* = 0.004) or high lipid exposure (*p* = 0.008) alone, and the combination resulted in ~22% less accretion at 180 min (*p* < 0.0001). The diabetic fuels did not lower the BODIPY MU C12 uptake across the full course of experiments. Instead, earlier in the assay, at 20 and 60 min, the high lipid- and combination-exposed BeWo had approximately three times the accumulation of BODIPY MU C12, consistent with a concentration-dependent early uptake. However, this initial rate of uptake was not sustained during the second (biphasic) uptake, so there were no differences in accretion at 180 min. The statistically significant differences in fuel-mediated whole-cell FA uptake at each time point are shown in Figure 5D. Our data show that diabetic fuels impair longer-chain FA uptake in BeWo.

### 2.6. Fatty Acid Partitioning into Lipid Droplets Varies by Fatty Acid Species and Fuel Exposures

To understand how diabetic fuels impact the lipid droplet partitioning of various FA species, we used particle analyses to measure the lipid droplet intensity separately and calculated the proportion of fluorescence in droplets compared to the whole cell. Representative images of droplet regions of interest (ROIs), shown in Figure 6A, demonstrate the methods used to threshold lipid droplets using BODIPY 505/515 counterstain, which clearly identifies lipid droplets containing neutral lipid (left) and colocalizes with BODIPY MU C12 (right). A similar threshold method was used to identify droplets containing BODIPY C12 with C16 (Appendix A). The proportion of BODIPY C12 and C16 in the lipid droplets increased steadily across time, whereas the proportion of BODIPY MU C12 in the droplets peaked earlier, at 60 min, before the proportion dropped back down as more whole-cell accumulation occurred during the second biphasic peak (Figure 6B). The effects of the diabetic fuels on the lipid droplet accumulation are shown in Figure 6C. Compared to controls, the proportions of BODIPY C12 in the lipid droplets was significantly less in the high glucose- and the combination-exposed BeWo at 180 min (*p* = 0.0329 and *p* = 0.0002, respectively), leaving a higher proportion in the cytoplasm or other organelles. High lipid-exposed BeWo had a higher proportion of BODIPY C12 partitioned in the lipid droplets compared to controls at 20 and 60 min (*p* = 0.0056 and 0.0013, respectively), but this was not sustained at later time points as whole-cell uptake continued. BODIPY C16 had the most significant fuel-mediated differences in lipid droplet partitioning. Very early, at 5 min, the high glucose and high lipid groups had at least double and the combination- exposed cells had triple the proportion of BODIPY C16 uptake in their droplet ROIs compared to controls (*p* = 0.0059 for high glucose, *p* = 0.0017 for high lipid, and *p* < 0.0001 for combination). However, as the whole-cell uptake of BODIPY C16 increased over time, the proportion within the droplets decreased so that, at 180 min, there was 38% less BODIPY C16 partitioned in the droplets in the high glucose-exposed cells (*p* = 0.0302) and 50% less in the combination-exposed cells (*p* = 0.0008) compared to controls. For BODIPY MU C12, the high glucose-exposed cells had 2.86 times more relative fluorescence in the droplets than controls at 120 min (*p* = 0.0310) and the combination-exposed cells had 1.6 times more than controls at 60 min (*p* = 0.0367). Lipid droplet partitioning of BODIPY MU C12 occurred but was not sustained at 180 min, as the proportion in the cytoplasm increased during the second phase of uptake. Overall, it was found that the proportion of FA partitioned to the lipid droplets over time varies by FA carbon length and saturation, and diabetic fuels modify subcellular stores.

### 2.7. High Lipid and Combination Exposures Alter the Number and Relative Area Occupied by BODIPY C16 Droplets over Time

The comparison of both lipid droplet count and the relative area of accumulation within the whole cell helped us delineate the fuel-mediated variations in the BODIPY C16-containing droplets (Figure 7A). Compared to controls, the high lipid-exposed BeWo had significantly more droplets at 5 and 60 min (*p* = 0.0048 and *p* = 0.0289, respectively) and the combination-exposed BeWo had significantly more droplets at 20 and 60 min (*p* = 0.0034 and *p* < 0.0001) (Figure 7B). In the images obtained after 60 min (at 120 and 180 min), the whole-cell BODIPY C16 uptake was high enough that we could not confidently threshold individual droplets utilizing our predetermined method (Figure 6A and Appendix A). Therefore, the total area of BODIPY C16 accumulation per cell was used to determine the impact of the diabetic fuels at those times (Figure 6C). While, at 20 min, the combination-exposed BeWo had more of the cell area taken up by C16-containing droplets than controls, by 60 min, there was no fuel-mediated difference, and by 180 min the combination-exposed BeWo had significantly less cell area taken up by BODIPY C16-containing droplets (*p* < 0.0001) (Figure 7C). Here, we show that diabetic fuels not only alter the proportion of BODIPY C16 stored in BeWo (Figure 6C), but also the number of lipid droplets (Figure 7B) and their relative abundance within the cell (Figure 7C). Specifically, high lipid conditions increase the number of C16-containing lipid droplets while high glucose conditions decrease the relative area of accumulation. The overall effect of the combination is more droplets with less BODIPY C16 accumulation over time.

### 2.8. BODIPY C16-Containing Lipid Droplets Move Centrally over Time, but Dynamics Are Altered by Diabetic Fuels

We also observed variable sub-cellular localization of the BODIPY C16-stained lipid droplets in the BeWo cells, with a perceived peripheral to central progression over time; therefore, we designed image analyses to map the relative distance of lipid droplets from the cell’s center over time (Figure 8A). Regardless of media exposure, the relative distance of BODIPY C16-stained droplets from the cell’s center decreased with time as BODIPY C16 droplets moved centrally (Figure 8B). A simple linear regression showed that the slope or rate of central movement did not differ between the media-exposure groups (*p* = 0.3757). However, at 20 min, the high lipid- and combination-exposed cells had droplets that were significantly closer to the cell’s center than the high glucose-exposed cells (*p* = 0.0292 and *p* = 0.0209, respectively), which suggests that they move centrally more quickly (Figure 8C,D). Using Pearson’s correlation, we tested whether the number of lipid droplets affected movement towards the center. There was no significant correlation between the number of droplets and the distance except for a positive relationship in the control and combination-exposed groups at 20 min (r = 0.3433, *p* = 0.0136 for control and r = 0.3459, *p* = 0.0112 for combination), which suggests that the cells with the most droplets tended to have clusters on the outer edges (Appendix A). Taken together, this indicates that BODIPY C16-containing droplets tend to move centrally over time, and the initial rate at which they do so is faster under high lipid conditions.

### 2.9. Primary Cytotrophoblasts Also Have Carbon Length-Dependent Fatty Acid Uptake That Diminishes After Syncytialization

Using the same fluorescently labeled long-chain FA analogues, BODIPY C12, C16, and MU C12, we used confocal microscopy to measure the FA uptake in primary human CTBs cultured for 12 h (Figure 9A) and for 96 h (Figure 9B). Like BeWo, the CTBs took up BODIPY C12 and MU C12 quickly and then the whole-cell uptake leveled off at 60 min, whereas the C16 uptake was slower and continued to increase over the course of three hours. At 96 h, primary trophoblasts clustered into a syncytium, which is more characteristic of syncytiotrophoblasts (Appendix A). These syncytialized trophoblasts had significantly less FA uptake than individual CTBs (Figure 9B). Specifically, the syncytialized trophoblasts had minimal uptake of BODIPY C12 and MU C12 species and accretion was observed to occur predominantly in lipid droplets while BODIPY C16 had more uniform uptake across the whole-cell and lipid droplets. Like BeWo, primary isolated cytotrophoblasts accrete longer-chain FA more slowly and have very little monounsaturated FA uptake. Overall, FA uptake is also much less after syncytialization.

## 3. Discussion

In this study, we tested the hypothesis that the combined exposure of excess glucose and lipids, as seen in GDM, impairs mitochondrial function and FA uptake in trophoblasts. Using BeWo, we showed that exposure to high glucose and high lipid conditions impairs mitochondrial respiration and cell viability. We used BeWo and primary cytotrophoblasts to establish species-specific differences in FA uptake, with both the amount and rate being dependent on the acyl chain length and degree of saturation. Specifically, we demonstrated that longer-chain FAs are accreted more slowly and that diabetic fuels further impair their uptake in BeWo. Additionally, we used a novel custom-designed methodology to show that diabetic fuels not only impair the overall uptake of longer-chain FAs, but also lipid droplet partitioning and dynamics. Collectively, our findings could explain why both placentas and infants exposed to GDM have LCPUFA deficiency despite there being more lipid availability.

While the standard treatment for GDM focuses on glucose control, increasing evidence suggests that exaggerated hyperlipidemia plays an important role in moderating consequences for both the pregnancy and the progeny. In this study, high lipid exposure compounded the effects of high glucose exposure, leading to mitochondrial dysfunction and trophoblast apoptosis (Figure 1). Regulated apoptosis is necessary for normal placental development [43], but how it impacts FA transport is unknown. Others have shown that in vitro exposure to excess individual FAs [44,45] or glucose [46] alone can cause apoptosis of placental cells. Excess apoptosis could influence placental development early in pregnancy, as well as the ability of trophoblasts to proliferate and ultimately differentiate to carry out the crucial functions across pregnancy. Indeed, hyperlipidemia is a recognized risk factor for both GDM and preeclampsia [47,48,49]. Our study adds evidence that hyperlipidemia impacts trophoblast fate, which can affect placental health to increase pregnancy-related consequences associated with GDM and other metabolic diseases.

Because FAs drive mitochondrial biogenesis and incite damage and mitophagy, it is important to study the effects of glucose and lipid overload on both the quantity and quality of mitochondria. Our findings show that the combination of high glucose and high lipid conditions generates more but less functional mitochondria (Figure 2 and Figure 3). Indeed, the respiratory capacity dropped significantly in combination-exposed trophoblasts despite an increase in mitochondrial abundance, which could explain the observed effects on both cell fate and function. Despite impaired respiration, we did not find a fuel-mediated difference in ATP production. It is possible that the ATP production appeared to be stable despite impaired respiration because we calculated the ATP levels using extracellular flux analysis data rather than directly measuring it. Other studies have shown that lipotoxicity impairs ATP production [50,51,52]. It is also possible that glycolytic BeWo can maintain ATP production with minimal glucose in the media, even in the face of impaired mitochondrial respiration. Regardless, our findings that FA uptake was disrupted even when ATP production was unchanged suggest that perturbations in metabolic homeostasis play a more significant role than we initially recognized. Understanding how mitochondrial dysfunction directly impacts FA uptake remains an important area for ongoing research.

To characterize the kinetics of cellular FA uptake over an extended period, we utilized saturated BODIPY-labeled FAs of different lengths and saturations. A difference of four carbons or the addition of a double bond dramatically affected the rates of FA accretion into BeWo and CTB cells (Figure 4 and Figure 9). To our knowledge, this is also the first study to quantify FA uptake in human trophoblasts over three hours and to assess the FA partitioning in lipid droplets (Figure 6). BODIPY C12, which mimics a 16-carbon, saturated FA (e.g., palmitate), had no fuel-mediated differences in whole-cell uptake. However, less BODIPY C12 was sequestered in the lipid droplets in high glucose- or combination-exposed BeWo. This localization pattern is consistent with more FA being available for export or deposition into cell membranes and organelles [53,54,55]. BODIPY C16, which mimics a longer, 20-carbon, saturated FA, had much slower uptake. Across three hours, the high glucose- and combination-exposed cells had less whole-cell uptake and lower proportions of BODIPY C16 in their lipid droplets. This pattern suggests an overall fuel-mediated decrease in trophoblast uptake and partitioning of the long-chain fatty acids we studied. If the acyl chain length is the primary driver in differences in fatty acid uptake, this could also explain the low LCPUFA levels in both GDM placenta and fetuses. We also found that compared to other FA species, monounsaturated (MU) BODIPY C12 had markedly lower whole-cell uptake with a biphasic pattern. A possible explanation for this pattern is that unsaturated FAs are taken up into trophoblasts in lower amounts and more slowly as an acyl-CoA is added to prevent export [56], then unsaturated FAs are temporarily partitioned in lipid droplets before being released back into the cytosolic free FA pool [54] for export to the fetus or use by the cell. Overall, the diabetic fuels did not change the uptake of BODIPY MU C12 across the three hours. However, the combination-exposed trophoblasts had a higher proportion of BODIPY MU C12 in their droplets at 60 min, suggesting earlier shuttling to the droplets, possibly to prevent oxidation by ROS generated from mitochondrial dysfunction. Whether multiple double bonds, as seen in LCPUFA, affect uptake requires further study.

Our novel methods also allowed us to characterize the lipid droplets’ count, size, and subcellular localization in BeWo up to 60 min (Figure 7 and Figure 8). While this approach does not address all of the complexities of the “lipid droplet life cycle” [54], it is important to study the effects of diabetic fuels on lipid droplet dynamics [53,55]. Here, we showed that high lipid exposure increased the number of BODIPY C16-containing droplets, but high glucose exposure decreased their relative area within the cell, so that the net effect of combination exposure is more droplets with less area of accumulation, suggesting that the droplets housed less BODIPY C16. While the identified droplets typically exhibited co-localized accumulation of all tested FAs, we cannot delineate the content of pre-existing droplets or stored non-fluorescent fatty acid species, which could explain discrepancies between the relative fluorescence in the droplets and the relative sizes of droplets. Lipid droplet characteristics vary based on cell type, fuel, and cellular metabolism [57]. One concept concerning lipid droplets is that larger droplets are generated to reduce lipotoxicity caused by lipid overload, which can come from excess uptake or poor metabolism [21,22], and that cells that accumulate more droplets not only have more reactive oxygen species production but may also be protecting neighboring cells from lipotoxicity [58,59]. In our study, we observed that diabetic fuel-exposed BeWo had a higher number of lipid droplet that were trafficked centrally more rapidly, which is consistent with this concept. Droplet interactions with other organelles are not well understood [54], but interactions with the endoplasmic reticulum, microtubules, and cytoskeleton are potentially important for cellular responses to lipotoxicity [54,60,61,62]. While this study uncovered interesting fuel-mediated changes in the subcellular movement of lipid droplets, further research is required to assess lipid droplet interactions with the endoplasmic reticulum, mitochondria, and the nucleus in varying fuel environments.

A strength of our study was measuring the baseline uptake of different fatty acid species in both BeWo cells and primary isolated cytotrophoblasts (CTBs). When tested 12 h after isolation, the FA uptake in the CTBs had a similar pattern of whole-cell FA uptake as BeWo (Appendix A). In addition to shared features between the CTBs and BeWo, including fatty acid transporters [63,64], glucose transporters [65,66,67], and the ability to take up non-esterified fatty acids [63], their similar kinetics of FA uptake lends support for using BeWo to measure fuel-mediated effects without introducing the confounding clinical factors that come from primary CTBs (Appendix A). We also assessed the FA uptake in primary CTBs cultured for more than 72 h, after which the CTBs spontaneous self-differentiated into syncytiotrophoblasts (SCT) [68]. We showed that syncytialization (Appendix A) led to a decrease in FA uptake. In line with previous reports [69], this study validated that the BODIPY C12 uptake in syncytialized cells is negligible in the first 20 min. However, using a longer time course, we showed that BODIPY C12 did eventually co-localize with the more abundant BODIPY C16-positive droplets. Our study affirms that the reported metabolic differences between CTBs and syncytiotrophoblasts may indeed influence FA uptake [40,70].

While our study had many strengths, it was not without limitations. We cannot predict the collective effect of high glucose and high lipid conditions on the placental function across a full-term pregnancy. Our high lipid condition exposed BeWo cells to a broad range of physiological lipids, which is a strength, but it prevents the ability to identify the effects of any individual FA species. This study did not assess the release of FAs for fetal accretion. Additionally, the approach of using primary CTBs is limited by the fact that they are not directly at the maternal–fetal interface; rather, they are the metabolic drivers of the placenta. Future work is also needed to evaluate the FA uptake in syncytiotrophoblasts and the relationships between CTB metabolism and FA processing in syncytiotrophoblasts. The technical limitations of this study include that the resolution of images and experimental time points limits the interpretation of intracellular lipid droplet dynamics in between select time points, which may have prevented us from capturing other findings in these remarkably dynamic structures [54]. Finally, we only assessed the effect of diabetic fuels on mitochondria abundance and respiration, which limited our ability to study other metabolic- or mitochondria-mediated mechanisms, including ROS production, which could drive variable FA uptake in trophoblasts. Future co-localization studies and the use of fluorescent analogs that measure metabolic homeostasis would bring more mechanistic insight.

## 4. Materials and Methods

### 4.1. Human Subjects

This study includes data and human primary isolated cytotrophoblasts (CTBs) from seven consenting participants enrolled in Sanford’s Feasibility of Cord Blood/Umbilical Cord Tissue Collection, Cryopreservation, Thawing, and Cellular Analysis and Placental Tissue/Cellular Analysis Study (STUDY00000571, PI: Baack). This observational, non-interventional study was approved by the Sanford Research Institutional Review Board in 2016 and follows Federal Regulatory Guidelines (Federal Register Vol. 46, No. 17, 27 January 1891, Part 56) and the Office of Human Research Protection regulations (45 CFR 46). Written, informed consent was obtained from all participants. Inclusion criteria include consenting donors between the ages of 18–45 who are pregnant and scheduled for a planned cesarean-section delivery at Sanford Health. Subjects who have known HIV/AIDs, hepatitis B, hepatitis C, or other blood-borne disease, have gestational age <34 weeks at delivery, placenta abruption, or chorioamnionitis or are unable to give an informed consent for reasons of incapacity, immaturity, adverse personal circumstances, or lack of autonomy were excluded. Participants were also excluded if their delivering physician felt that collection of the cord blood, tissue, or placenta could affect the health of the mother or infant. All consenting parents in this study are female, although the placenta’s “sex” mirrors that of the fetus and both male and female placentas were used for CTB isolation. Demographics for participants in this study are detailed in Table 1.

### 4.2. Primary Cytotrophoblast Isolation and Culture

Placental sections were collected immediately after delivery and placed in sterile containers, then transported for tissue processing 1–2 h after delivery. Villous tissue was removed from the vessels and connective tissue using the blunt end of a scalpel. Approximately 50 g of villous placenta tissue was minced and subjected to three sequential 25 min (37 °C) digestions in 1.5 mg/mL trypsin (Thermo Fisher Scientific, Waltham, MA, USA, 27250018), 1 mg/mL dispase II (Gibco, Waltham, MA, USA 17-105-041), and 0.1 mg/mL DNAse (Sigma Aldrich, St. Louis, MO, USA, DN25-5G). The mixture was filtered through a 100 µm cell strainer, then 3 mL FBS was layered beneath the mixture in a conical vial. This was pelleted, then resuspended in 5 ml of complete Minimum Essential Medium α (MEMα (Corning, Corning, NY, USA 10-022-CV), 10% fetal bovine serum (FBS, Corning, 35-010-CF), 100 U/mL penicillin + 100 µg/mL streptomycin (P/S, Thermo Fisher Scientific, 15-140-122)) before purification using a Percoll density gradient (60% to 20%) and centrifugation at 1250 rcf for 25 min at room temperature. CTBs were plated at a density of 1.0 × 10^5^ cells/dish in gelatin-coated 35 mm imaging dishes and cultured in complete MEMα. Medium was replaced every 24 h and cells were used for fatty acid uptake analyses at 12 and 96 h, as detailed below.

### 4.3. Trophoblast Cell Line and Culture Conditions

BeWo, a human choriocarcinoma cell line, was purchased from the American Type Culture Collection (ATCC, Rockville, MD, USA, CCL-98). Cells were cultured in F12K media (ATCC, 30-2004) supplemented with 10% FBS (Corning, 35-010-CF) and 1% P/S (Thermo Fisher Scientific, 15-140-122). All cells were utilized between passages 10–20 and were maintained in sterile conditions at 37 °C in 5% CO_2_.

BeWo cells were routinely cultured in basal media (F12K + FBS) as recommended [72]. BeWo cells were also cultured in high glucose (HG), high lipid (HL), and combination conditions to examine individual and combined effects of diabetic fuels on cell fate, metabolism, and FA uptake. When supplemented with 10% FBS, the F12K media contains 7.5 mM glucose, which is a relatively physiologic glucose level. For HG culture conditions, the basal medium was supplemented to contain 25 mM glucose to approximate uncontrolled diabetic conditions and replicate what has been done in previous studies [36,73,74]. For HL culture conditions, basal medium was supplemented with 1% Chemically Defined Lipid Concentrate (CDLC, Thermo Fisher Scientific, 11905031), a concentrated lipid emulsion designed for mammalian cell culture. CDLC contains a defined mixture of cholesterol and FAs including myristic (14:0), palmitic (16:0), palmitoleic (16:1), stearic (18:0), oleic (18:1), linoleic (18:2), linolenic (18:3), arachidonic (20:4), α-tocopherol, and pluronic F-68, the latter of which is a non-ionic surfactant. There are no reported lipids in F12K media and the FBS used for these experiments had a reported 7 mg/dL of low-density lipoproteins (LDLs), 10 mg/dL of high-density lipoproteins (HDL), and 63 mg/dL of triglyceride (TG). They additionally report insulin to be 9.53 µIU/mL (batch-specific documentation). To approximate the exaggerated physiologic hyperlipidemia associated with GDM [75,76], BeWo cells were cultured in basal media + 1:100 CDLC, which yields approximately 2-fold increase in lipid exposure. To validate the exposure, non-esterified fatty acid (NEFA) levels in media were measured using a Wako HR Series NEFA kit (Fujifilm, Lexington, MA, USA). By our measurements, basal media has 0.06 meq/L of NEFA and 1:100 CDLC supplemented media has 0.11 mEq/L (Appendix A). This correlates with a 1.82-fold increase in NEFAs, reflecting the relative fold change in circulating lipids in GDM rather than absolute level of circulating lipids [16]. The combination culture conditions include 25 mM glucose and 1% CDLC.

For fuel exposure experiments, corresponding cell medium was replenished every 24 h for at least 72 h, then BeWo cells were collected for cell fate, protein abundance, Seahorse XF analyses, and fatty acid uptake experiments as detailed below.

### 4.4. Cell Fate Assays

Equal numbers of cells were seeded on 100 mm cell culture dishes in F12K + 10% FBS + 1% P/S and allowed to adhere before treatment with HG, HL, or combination media for at least 72 h. For growth curve, 4.0 × 10^5^ cells were seeded into a 24-well plate, three wells per media group. At 24, 48, 72, and 96 h, adherent cells were harvested with trypsin and counted. Doubling time, or the time it takes for a cell population to double in number, was calculated with the cells counted at 24 h and the cells counted at 96 h. The fold change in cells counted between 24 h and 96 h was calculated by the difference in counts divided by the cell counts at 96 h. To quantify apoptosis, both cultured cells and supernatant were harvested at 72 h (to collect dead/floating cells) and washed with ice-cold PBS prior to staining using APC Annexin V Apoptosis Detection Kit (BioLegend, San Diego, CA, USA) according to manufacturer’s protocol. Samples were analyzed by flow cytometry (BD Biosciences Fortessa, Franklin Lakes, NJ, USA), using control samples to set gating. Annexin V+propidium iodide(PI)-stained apoptotic cells, PI-only-stained necrotic cells, and non-stained live cells were counted and group comparisons were made.

### 4.5. Western Blot Protein Abundance Analyses

The BeWo cells were harvested and gels and blots were generated and imaged in parallel to minimize variation between biological replicates, as previously described [77]. Membranes were washed before imaging with Luminata Forte HRP Chemiluminescence Substrate (Thermo Fisher Scientific) and LI-COR Odyssey XF Imager with Image Studio™ Lite Version 5.2 (LI-COR, Lincoln, NE, USA). Densitometry was performed using the aforementioned Image Studio software. Beta-actin served as the reference protein and each band represented a biological replicate. For quantification, each band was normalized to the average of the respective control. The following antibodies were used and diluted in 1:1 EveryBlot Blocking Buffer:TBS-T: anti-TOM20 (1:1000, Cell Signaling Technology, Danvers, MA, USA #42406), anti-VDAC (1:1000, Cell Signaling Technology #4661), anti-β-actin (HRP-conjugated) (1:1000, Cell Signaling Technology #5125), anti-rabbit IgG-HRP (1:5000, SouthernBiotech, Birmingham, AL, USA #4030-05), and anti-mouse IgG-HRP (1:5000, SouthernBiotech #1031-05). Full representative blots can be found in Appendix A.

### 4.6. Seahorse XF Analyses

Following culture conditions described above, BeWo cells were plated at 1.5 × 10^5^ cells/well in Seahorse XF24 V7PS plates and real-time cellular bioenergetics were measured using a mito stress test (MST), a glycolysis stress test (GST), and the seahorse XFe analyzer (Agilent Technologies, Santa Clara, CA, USA). For the MST, oxygen consumption rate (OCR) was measured at baseline and following injections of validated doses of oligomycin (1 uM), FCCP (0.3 uM), and rotenone plus antimycin A (0.5 uM each). OCR measurements were normalized to cell number using Hoechst fluorescence at the end of the MST. Basal respiration, maximal respiration, spare respiratory capacity, proton leak, proton efflux rate, and the MST contribution to ATP production were calculated as previously detailed [78,79,80,81]. For the GST, cells were cultured in glucose-depleted assay media for 1 h of degassing before extracellular acidification rate (ECAR) was measured at baseline and after sequential injections of glucose (10 mM), oligomycin (1 uM), and 2-deoxyglucose (50 mM). ECAR measurements were normalized to cell number using Hoechst fluorescence at the end of the GST. Basal and maximal glycolytic rate, glycolytic reserve, and non-glycolytic acidification were calculated as previously detailed [78,79,80,81,82].

### 4.7. Fatty Acid Uptake Analyses

#### 4.7.1. BODIPY Labeling

BODIPY-labeled FAs have been used to study FA uptake and esterification in multiple organ models [69,83,84,85,86,87,88,89], including placental explants [69]. BODIPY C12 and C16 are 12- and 16-carbon chain-length saturated FAs linked to the fluorophore BODIPY (4,4-difluoro-3a,4a- diaza-s-indacene) to biologically resemble and behave like C:16- and C:20-carbon length FAs with regard to transport and metabolism in cells. Thus, BODIPY-labeled FAs are effective markers for lipid trafficking. In addition to labeling BeWo with manufactured BODIPY C12 (Invitrogen, Waltham, MA, USA, D3822) and BODIPY C16 (Invitrogen, D3821), we were kindly gifted an aliquot of a monounsaturated version of BODIPY C12 that was developed by Dr. Summer Gibbs for Dr. Kent Thornburg, who’s lab also studies FA uptake and metabolism in trophoblasts and cardiomyocytes [69,83,84,85].

For these studies, 1 mg/mL stocks of BODIPY C12, BODIPIY C16, and the monounsaturated BODIPY C12 (MU C12) in DMSO were used alongside BODIPY 505/515 (Invitrogen, D3921) to counter-stain neutral lipids during experiments using MU C12. Viable cells were counted using Trypan blue and live-cells were seeded onto gelatin-coated 35 mm imaging dishes at a density of 5.0 × 10^4^ and incubated overnight, 12–18 h. BeWo cells were ultimately stained with 1 µg/mL of the BODIPY FAs, 1 µg/mL Hoechst, and 20 µg/mL BODIPY 505/515. CTBs took up BODIPY more readily, exceeding threshold parameters at the same dose. Therefore, CTBs were stained with 0.25 µg/mL of the BODIPY FAs (1/4 concentration used in BeWo), 1 µg/mL Hoechst, and 10 µg/mL BODIPY 505/515.

#### 4.7.2. Live-Cell Imaging & Time Points

After incubation with BODIPY C12, C16, or MU C12 for 5, 20, 60, 120, or 180 min, cells were washed with prewarmed PBS three times, then incubated for 20 min with Hoechst and BODIPY 505/515 counterstains for MU C12 uptake experiments. Images were captured using a Nikon A1 TIRF Ti-Eclipse inverted confocal microscope (Nikon Instruments Inc., Melville, NY, USA) equipped with a live cell chamber (37 °C with humidified 5% CO_2_), and NIS-Elements AR image acquisition software (Version 5.42.06, Nikon). Individual cells were selected for imaging in a standardized manner by a single technician based on morphology and nuclear stain quality at the start of imaging, with 10–15 cells (technical replicates) selected across the dish per time point per experiment, with three experiments performed. Notably, images for analyses were captured at the slice beneath the nucleus to limit variation in nucleus size (and consequential lack of fluorescent FAs). All images were captured at 60× magnification and 1024 × 1024 pixels with 1.1 µs pixel dwell (6.27 s/frame). The same laser power and acquisition settings were used to directly compare red species (BODIPY C12 and MU C12); however, each of these settings was individually optimized for each FA species to quantify the effect of media on uptake.

### 4.8. Image Analysis

Images obtained by live-cell confocal microscopy were analyzed using Fiji software, ImageJ Version 1.54f [90] to quantify whole-cell and lipid droplet FA uptake over time, as well as lipid droplet dynamics.

#### 4.8.1. Whole-Cell Fluorescence

Cell ROIs were automatically segmented using the Triangle method of threshold and particle analysis. If automatic threshold method did not generate continuous cell boundaries automatically, partial freehand selection allowed the cell boundary to be manually traced. Areas of whole-cell ROIs were measured (µm^2^) and intracellular C12, C16, and MU-C12 FA uptake was calculated using the ROI’s mean fluorescence per area (fluorescent intensity (arbitrary units)/µm^2^).

#### 4.8.2. Lipid Droplet Partitioning

Within each cell’s ROI, particle analysis of the green channel was segmented using the Shanbhag method to identify lipid-laden droplets within cytoplasm. The relative fluorescent intensity of BODIPY FAs was quantified using Integrated Density (IntDen) in ImageJ to delineate the fluorescence within droplet ROIs compared to whole-cell ROIs. The proportion of individual FAs within droplets per cell was calculated as the sum of the IntDen measured by droplet ROIs divided by the IntDen measured by the cell ROI. Proportion of total fluorescence in “non-droplets” for each cell was the 1-(proportion of fluorescence in droplets).

#### 4.8.3. BODIPY C16 Area of Accumulation

Droplet ROIs were identified as described above (4.8.2). The sum of this measured area per the cell ROI area represented relative area of accumulation.

#### 4.8.4. Lipid Droplet Localization

Lipid droplets are very dynamic and mobile within the cell. After observing significant fuel-mediated differences in BODIPY C16 uptake, we calculated lipid droplet number and sub-cellular localization of intracellular BODIPY C16-labeled droplets over 60 min. The ImageJ Cell Counter plugin was utilized as shown in Figure 8 to generate: (1) three markers approximating the center of the cell, (2) three markers approximating the outer cell boundary, and (3) markers on every visible BODIPY C16 droplet within the cell. The ImageJ Measure function exported markers by their type as well as location in coordinates in microns, per the image scale, on an arbitrary x,y axis. From these coordinates, the following were calculated: (1) average center point, (2) distance between outer cell boundary and average center (the average of these points being the average cell radius), and (3) distance between each droplet and the average center point normalized to average cell radius, and reported this as relative distance across cell radius. A Pearson correlation with 95% CI was used to determine whether there was a significant relationship between droplet number and the relative distance across the cell radius. If a significant correlation was found, a post-hoc linear regression analysis was used to determine if the slope was non-zero, which would reflect whether the number of droplets influenced their relative distance across cell radius.

### 4.9. Statistical Analysis

All statistical analyses were conducted using GraphPad Prism Version 9.4.0. Unless otherwise specified, averages are reported as the mean ± standard error of the mean (SEM) with individual data points as indicated on figures. Statistical parameters, including experimental replicates (*n*), are detailed in results and figure legends. One-way ANOVA with Tukey’s multiple comparisons test was used to test fuel-mediated effects on cell fate, cellular bioenergetics, protein expression, FA uptake, and lipid droplet dynamics. Statistical significance was set at *p* < 0.05 for all cases.

## 5. Conclusions

In conclusion, the combination of excess glucose and lipids impairs mitochondrial function and inhibits the uptake of FAs with longer acyl chains, like BODIPY C16, in human trophoblasts. Our evidence suggests that the additional lipid exposure on BeWo cells not only affects the metabolism and viability of placental cells but also impairs their ability to take up crucial long-chain fatty acids that are deficient in placentas and infants of GDM pregnancies. As the role of lipids in the pathogenesis of GDM is more appreciated and investigated, the management of parental lipid levels, in addition to blood glucose, may aid in both the development and metabolic function of the placenta in order to transportthe fuels that influence the growth and metabolic programming of the baby.

## Figures and Tables

**Figure 1 ijms-25-11534-f001:**
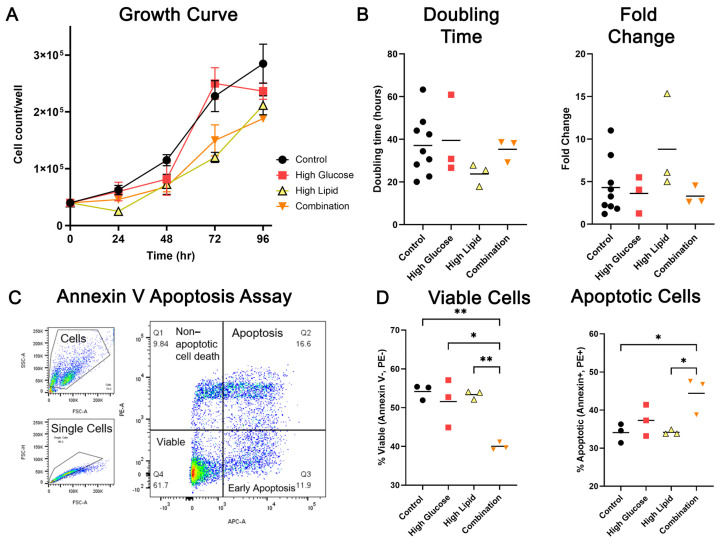
BeWo growth and viability in high glucose, high lipid, and combined conditions. BeWo cells were cultured in control, high glucose, high lipid, and combination media for 72 h and then uniformly plated to 24-well plates and cultured for 96 h in respective media. Daily cell counts were used to estimate growth over time (**A**) and calculate doubling time and fold change (**B**) from 24 h to 96 h. An apoptosis assay using flow cytometry was used to quantify APC-Annexin V (APC-A)-and PE-propidium iodide (PE-A)-tagged cells (**C**) to identify the percent of viable and apoptotic BeWo following 96 h of media exposure (**D**). *n* = 3/group; * *p* < 0.05, ** *p* < 0.01 by one-way ANOVA with Tukey’s multiple comparison test.

**Figure 2 ijms-25-11534-f002:**
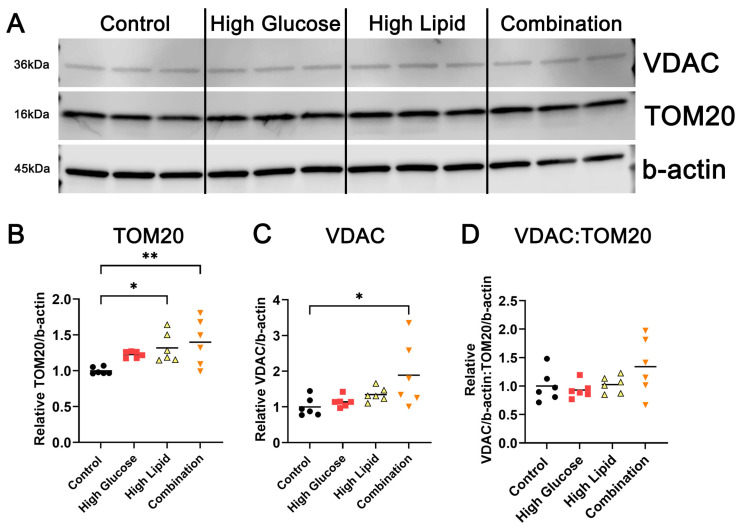
Mitochondrial protein abundance in high glucose-, high lipid-, and combination-exposed BeWo. Representative western blot (**A**), relative abundance (**B**,**C**), and ratio (**D**) of mitochondrial proteins TOM20 and VDAC in BeWo lysate. Densitometry was normalized to the average of controls on each well’s respective blot (*n* = 6/exposure group). * *p* < 0.05, ** *p* < 0.01 by one-way ANOVA with Tukey’s multiple comparison test. See full, unedited blots in Appendix A.

**Figure 3 ijms-25-11534-f003:**
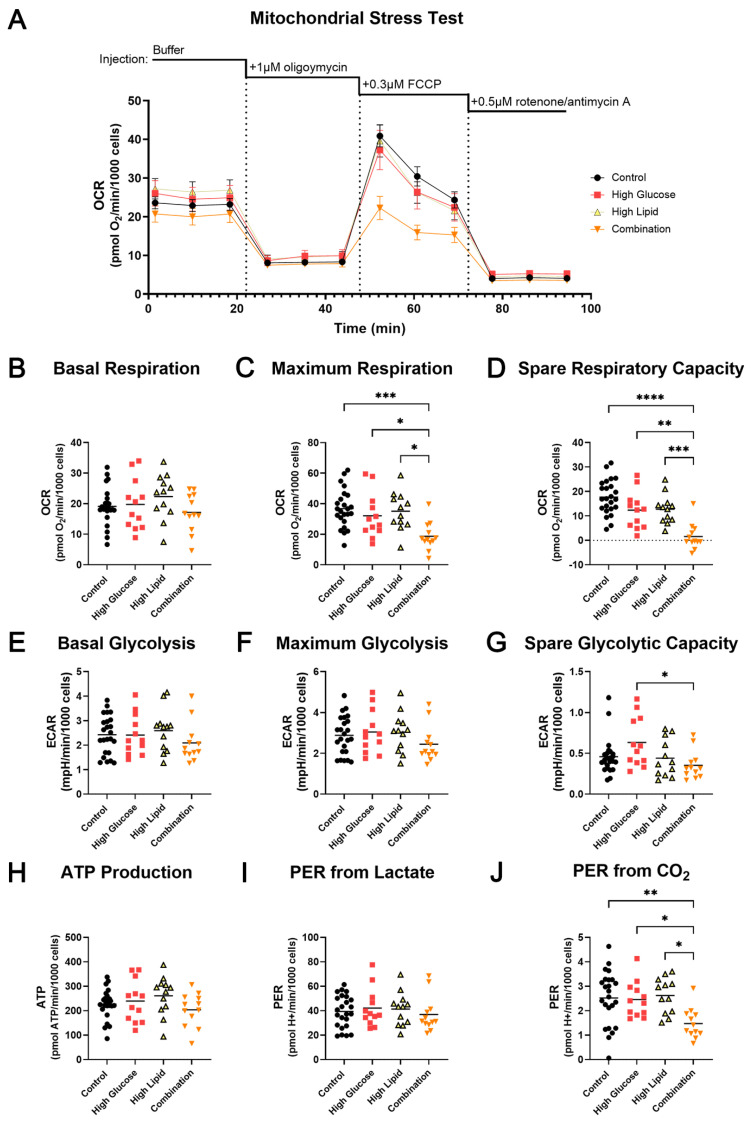
Cellular bioenergetics of control, high glucose-, high lipid-, and combination-exposed BeWo. Average oxygen consumption rate (OCR), which estimates cellular respiration, is shown as a trace across a mitochondrial stress test (**A**) and comparisons of average basal respiration (**B**), maximum respiration (**C**), and spare respiratory capacity (**D**) by group in BeWo cultured in control, high glucose, high lipid, and combination media. Average extracellular acidification (ECAR) estimates are shown for basal glycolysis (**E**), maximal glycolysis (**F**), and spare glycolytic capacity (**G**) by group. OCR and ECAR were used to calculate ATP production (**H**) and proton efflux rate (PER) leading to lactate production (anaerobic glycolysis) (**I**) and CO_2_ (aerobic glycolysis) (**J**). Values are mean ± SEM (**A**), and individual values from experimental replicates (**B**–**G**) and calculated values (**H**–**J**) are shown with the line representing the mean. *n* = 12–24/group. * *p* < 0.05, ** *p* < 0.01, *** *p* < 0.001, **** *p* < 0.0001 by one-way ANOVA with Tukey’s multiple comparison test.

**Figure 4 ijms-25-11534-f004:**
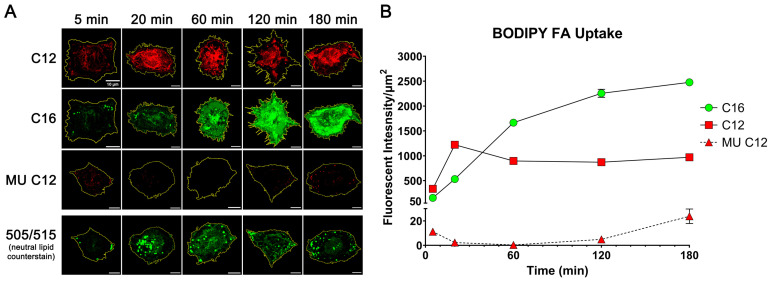
Fatty acid (FA) uptake by carbon length and saturation. Representative images of BeWo with whole-cell regions of interest (ROIs) were taken by confocal live-cell imaging at 5, 20, 60, 120, and 180 min after adding BODIPY C12 (red), BODIPY C16 (green), and monounsaturated BODIPY C12 (MU C12, red) to media (**A**). BODIPY 505/515 neutral lipid counterstain was used to validate that lipid uptake occurred in BODIPY MU C12 experiments. The average relative fluorescent intensities were plotted over time to assess variation in kinetics (**B**). *n* = 3 biological replicates/group with 41–55 cells/group/time point imaged and analyzed. Values are mean ± SEM.

**Figure 5 ijms-25-11534-f005:**
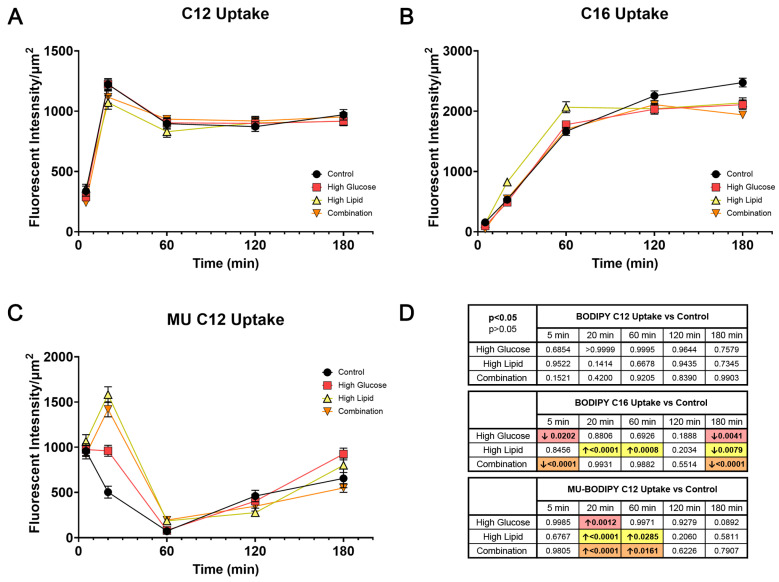
Whole-cell fatty acid uptake in controls and high glucose-, high lipid-, and combination-exposed BeWo over time. Each media group’s whole-cell uptake of BODIPY C12 (**A**), BODIPY C16 (**B**), and BODIPY MU C12 (**C**) are shown over 180 min, and group comparisons demonstrate time- and FA-specific differences between exposure groups (**D**). *p* values and arrow noting direction of change for statistical significance compared to control uptake across time points are shown (**D**). *n* = 3/exposure group with 41–55 cells/group/time point analyzed. Values are mean ± SEM. Significant differences from control *p* < 0.05 by one-way ANOVA with Tukey’s multiple comparison test.

**Figure 6 ijms-25-11534-f006:**
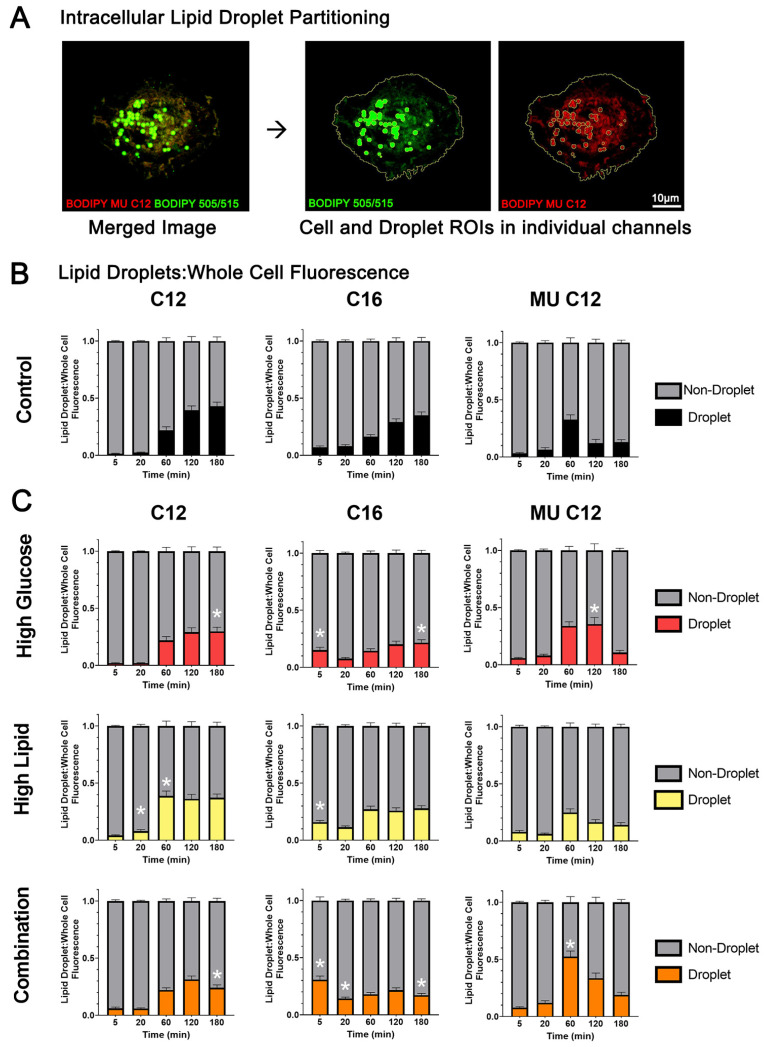
Effects of high glucose, high lipid, and combination exposure on the proportion of FA species in BeWo lipid droplets over time. Droplets were identified using the green channel fluorescence, BODIPY 505/515 (shown in A) or BODIPY C16 depending on the experimental design. Droplets were segmented using ImageJ particle analysis, as represented by BODIPY 505/515 (green) and BODIPY MU C12 (red) in BeWo imaged at 20 min (**A**). The proportion of droplet to whole-cell intensity was calculated, where 1 is the total fluorescence in the cell. This estimate of lipid droplet partitioning of individual FAs (BODIPY C12, C16, and MU C12) is illustrated over time in controls (**B**) and high glucose-, high lipid-, and combination-exposed BeWo (**C**). *n* = 3/exposure group with 41–55 cells/group/time point analyzed. Values are mean ± SEM. Significant differences (*p* < 0.05) from the control group at each time point by one-way ANOVA with Tukey’s multiple comparison test are indicated with the white asterisk (*) within the column.

**Figure 7 ijms-25-11534-f007:**
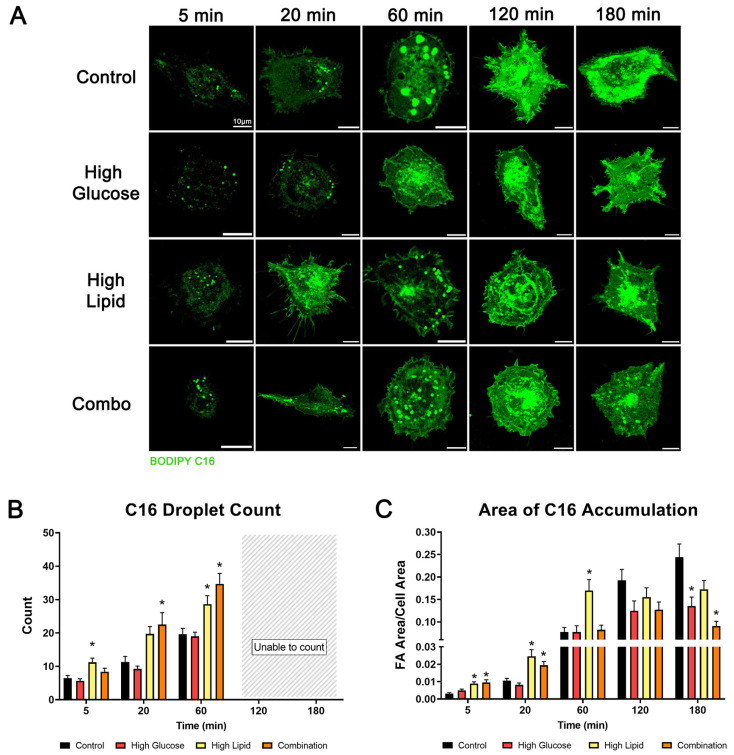
Representative images of BODIPY C16 lipid droplets in BeWo highlight the variations in amount and area occupied by lipid droplets (**A**). Average lipid droplet counts (**B**) and relative areas of BODIPY C16 accumulation per cell area (**C**) are represented by bar graphs per group, over time. *n* = 3/exposure group with 41–55 cells/group/time point. Values are mean ± SEM. Significant differences from control: * *p* < 0.05 by one-way ANOVA with Tukey’s multiple comparison test.

**Figure 8 ijms-25-11534-f008:**
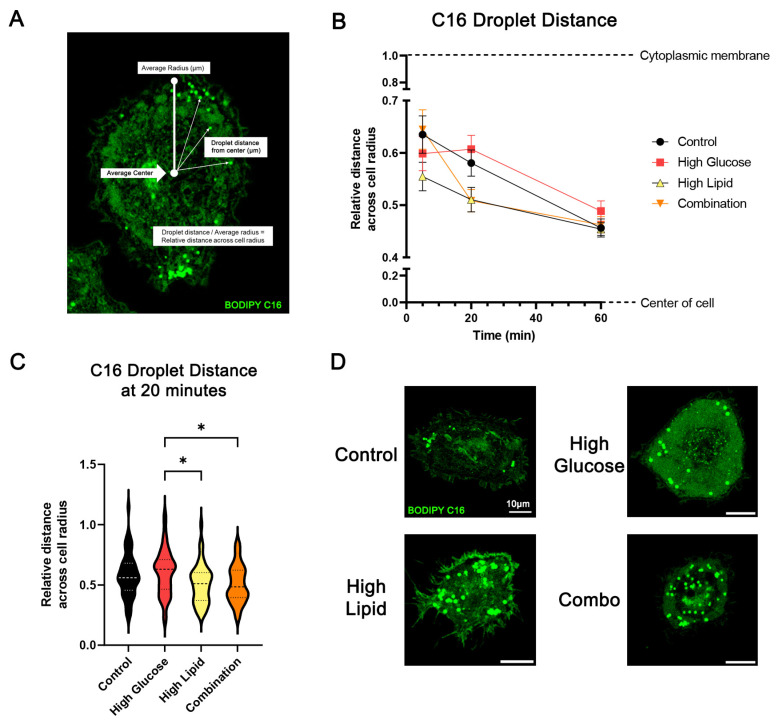
BODIPY C16-containing lipid dynamics over time. Strategy for determining localization of lipid droplets based on their distance from the center of the cell relative to the average cell radius (**A**) and BODIPY C16 droplet distance (**B**,**C**). Representative images of BODIPY C16 droplet distribution at 20 min in control, high glucose, high lipid, and combination media groups (**D**). *n* = 3/exposure with 41–55 cells/group/time point. Values are mean ± SEM. * *p* < 0.05 by one-way ANOVA with Tukey’s multiple comparisons test.

**Figure 9 ijms-25-11534-f009:**
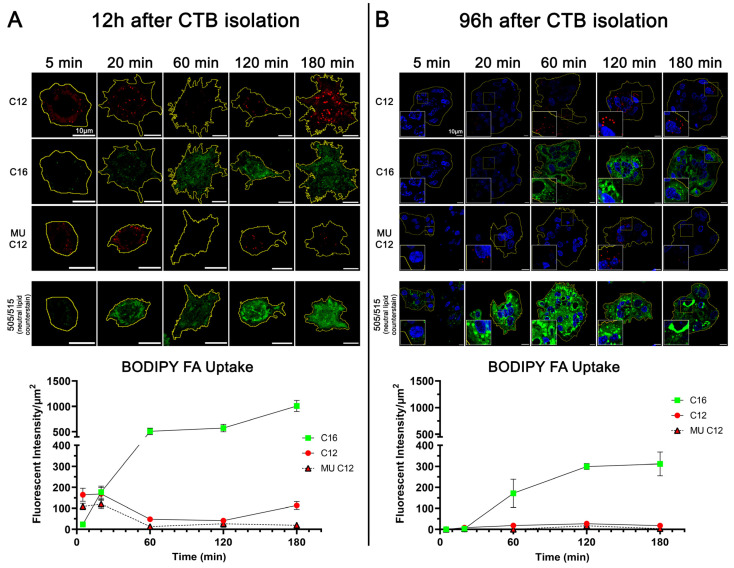
Fatty acid uptake in primary human trophoblasts. Representative images and relative fluorescence for each BODIPY FA take up in primary isolated cytotrophoblasts 12 h after isolation (**A**) and 96 h after isolation, whereby they have formed a syncytium (**B**). *n* = 7 patients, 10 cells measured per time point per patient. Values are mean ± SEM.

**Table 1 ijms-25-11534-t001:** Study Demographics.

	Mean ± SD (*n* = 7)	Media (IQ Range)
Age (year)	33.14 ± 6.20	35 (26–37)
Pre-pregnancy weight (kg)	97.79 ± 14.91	101.6 (83.50–110.7)
Pre-pregnancy body mass index (BMI, kg/m^2^)	35.90 ± 6.71	37.10 (28.81–42.56)
Gravidity	3.14 ± 1.46	2 (2–5)
Parity	2.57 ± 0.98	2 (2–4)
Weight gain (kg)	6.10 ± 4.96	4.1 (2.90–11.40)
Gestational Age (weeks)	38.45 ± 0.83	39 (37.43–39.28)
Birth weight (g)	3.44 ± 0.46	3.42 (3.09–3.86)
Small for gestational age (%)	0/7 (0%)	
Appropriate for gestational age (%)	5/7 (71%)	
Large for gestational age (%)	2/7 (29%)	1/2 with gestational diabetes mellitus (GDM) 1/2 with obesity 2/2 with pre-pregnancy BMI > 40 kg/m^2^
Male:Female	4:3	
Pre-pregnancy obesity ^1^ (%)	5/7 (71%)	
GDM (%)	2/7 (29%)	2/2 Type A2 2/2 treated with insulin and metformin 2/2 with obesity 2/2 with weight gain less than recommended per BMI [71]
Preeclampsia or chronic hypertension (%)	2/7 (29%)	2/2 with GDM 2/2 with obesity
Race (%)	7/7 (100%) Caucasian	
Ethnicity (%)	7/7 (100%) Non-Hispanic	

^1^ Obesity: BMI > 30 kg/m^2^.

## Data Availability

The raw data supporting the conclusions of this article will be made available by the authors on request.

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
