# Peer review of "Gestational Diabetes-like Fuels Impair Mitochondrial Function and Long-Chain Fatty Acid Uptake in Human Trophoblasts"

_ijms, 2024, doi:10.3390/ijms252111534_

Round 1

Reviewer 1 Report

Comments and Suggestions for Authors

The present study evaluated how metabolic fuels resembling those seen in gestational diabetes affect trophoblast mitochondrial function. The study is interesting and use of both cell line and primary trophoblast adds to its strength. I have the following comments that need to be addressed before the study can be published.

1. The discussion reads too long and can be edited to be concise. The authors are over-reaching/interpreting their observations eg. line 422-424.

2. In there a reasoning being why DHA was not used in the high lipid mixture?

3. In figure 2, what is the relevance of the VDAC: TOM20 ratio?

4. Since the combination exposure reduced viability and increased apoptosis in BeWo cells, could this account for the decline in maximum respiration and spare capacity seen in Figure 3? Were the exposure time and concentrations same for Figure2 /3 or did the authors reduce the exposure time to account for this variable?

5.  The citations/references on lines 181,182 section 2.4 are out of format

6. Figure 6 shows that there wasn’t much difference in droplet accumulation of C16 in high lipid exposure. How does this then corelate to the increased droplet numbers/accumulation in Figure 7? The authors again in discussion claim that there is increased c16 droplets but its not clear how they are concluding this? The figures 6 and 7 need to be clarified further and maybe together as currently they are not easy to understand.

7.  Since the manuscript deals with BeW’o’s and primary CTB, I would recommend using BeWo’s in the result section titles in place of trophoblast.

 8. The Figure 9 experiments were done in CTB pre and post fusion (representing STB) and show drastic differences in uptake. Most of the experiments are done in un-syncytialized BeWo’s which would represent CTB. In-vivo, CTB are not the cells responsible for direct uptake of fuels from the maternal system. Can the authors comment on this and put their results in this context in the discussion?

Reviewer 2 Report

Comments and Suggestions for Authors

This study revealed that excessive glucose and lipids impair mitochondrial function and modulate fatty acid uptake in trophoblasts. The authors provide analysis of cell viability and mitochondrial respiration in BeWo cells treated with high glucose, high lipid, and their combinations. The authors also analyzed the fraction of fatty acids in lipid droplets according to carbon length and saturation using BODIPY labeling. The study also reconfirmed the fatty acid uptake pattern in primary human trophoblasts. This study provides extensive experimental evidence for the detrimental effects of excessive glucose and lipids in human trophoblasts in GDM, and suggests the importance of maternal metabolic management. In the manuscript, the authors described the details in a way that readers can easily understand. I believe that this manuscript could be improved by considering the following minor points.

-       In the Results section, briefly describing the implications and conclusions of the content of that section at the end of each paragraph will help readers recall the implications of each experiment.

-       I recommend that the acknowledgment of gifted materials be placed in a separate section rather than in the main text, as in line 374.

Reviewer 3 Report

Comments and Suggestions for Authors

In this study authors evaluated the effects of high glucose, high lipid, and the combination on trophoblast growth, viability, mitochondrial bioenergetics, BODIPY-labeled fatty acid (FA) uptake, and lipid droplet dynamics. Authors found that the addition of four carbons or one double bond to FA acyl chains altered the uptake in both BeWo and primary isolated cytotrophoblasts (CTB). Uptake was further impacted by media exposure. Combination-exposed trophoblasts had more mitochondrial protein, but impaired maximal and spare respiratory capacity and increased apoptosis. Moreover, combination exposed trophoblasts had unimpaired uptake of BODIPY C12.

Although the manuscript and topic is interesting, the manuscript presents several points that deserve to be improved. In particular:

Lines 32-33: it deserves to be pointed that GDM is also a chronic inflammatory disease (PMID: 36359548). This is an important point to highlight since inflammation induces vascular damage (PMID: 37443812) and favors several post-partum complications found in this disease. 

Line 94: BeWo ell line is not an immortalized cell line (like HTR8/SVneo cells) but a choriocarcinoma cell line. 

Figure 2A: Molecular weights must be added. Moreover, VDAC blot exposion is very low and is not representative of the densitometric analysis reported in figure 2 B. 

4.5 Western Blot Protein Abundance Analyses: authors must add the product code of the primary antibodies used

In order to prove syncytialization occurrence, authors must show specific markers like E-cadherin, hCG beta or other. Moreover, spontaneous syncytialization occurrence is very low. For this reason, syncytialization is induced by forskolin treatment. 

Reviewer 4 Report

Comments and Suggestions for Authors

The aim of this study is to evaluate the effect of high levels of glucose, lipids or both on trophoblast growth, viability, mitochondrial bioenergetics, fatty acid uptake and lipid droplet dynamics. It was observed that the addition of carbons or double bonds to fatty acids affects the uptake in the cell cultures studied (BeWo and CTB) and how this effect is increased by exposure to the media studied. They also observed a higher mitochondrial protein content in trophoblasts exposed to the combination of media, but both the maximum respiratory capacity and the mitochondrial respiratory reserve were altered, as well as mitochondrial viability as a consequence of apoptosis. Finally, an altered uptake of C12 and C16 and alterations in lipid droplets in trophoblasts were observed. The conclusion obtained by these authors is that the results found can provide new data to explain the alterations observed in the placental transport of fatty acids in GDM and LCPUFA deficiency.

This is an article of scientific interest, which aims to provide new data necessary for understanding how excess energy nutrients during gestational diabetes can affect placental transport of fatty acids and the behavior of lipid droplets.

Although the abstract shows the most relevant results, I think it would be necessary to indicate, even briefly, whether there are differences between the different media, given that it only names the combination of both.

The introduction is clear, easy to read and justifies the objective pursued. It justifies why it uses two types of cell lines, one more stable and independent of pregnancy conditions and another more conditioned by pregnancy parameters and with less temporal stability, although it does not indicate what it is looking for with the use of these two lines, that is, would one have been sufficient or were both necessary?

Material and methods

Were the isolated primary cytotrophoblasts subjected to the same experiments as BeWo or not? Reading the material and methods, it seems that they were only isolated and observed by microscopy at 12 and 96 hours. This should be clarified, just as it has been done with the BeWo cells. I think that the material and methods should clarify what determinations are going to be made with this type of cells and, in addition, it should indicate, in some section, what is sought with the determinations made, that is, what the determinations in these cells really contribute to what was observed with the other cell line.

I think that without this information, obtaining these cells and the only determination made does not provide information of interest to this work.

Discussión

Page 12, lines 323-326: Could the authors clarify what it means that both CTB and BeWo present a unique uptake pattern, is it not shown in any other cell type?: “We used both BeWo and primary isolated CTB to show that human trophoblasts have unique FA uptake patterns based on acyl chain length and saturation and confirmed that FA uptake in primary isolated CTB is more similar to BeWo than to its differentiated counterpart, syncytiotrophoblasts”

Page 14, line 437: “One of the strengths of our study was carrying out these experiments in both BeWo cells and primary isolated cytotrophoblasts (CTB)”. This sentence is only partially true, since the vast majority of the experiments have been carried out only in BeWo.

Page 14, line 449: “we found significant variation among CTBs included in this study (Figure S5)”. This figure shows data from BeWo.

Overall, I consider the article to be of great scientific interest and provides interesting data related to mitochondrial damage in GDM, as well as information related to lipid uptake in placental cells of interest. There are small comments that would be interesting to resolve, including the need to perform some studies in CTB. The justification is related to demonstrating a similar pattern of lipid uptake between BeWo and CTB, although at no point does it indicate that these patterns were or could be different and this verification will need to be carried out. It is also related to seeing how lipid uptake decreases in the differentiation of CTB to SCT, although I do not believe that is an objective of this study.

Minors

Clarify the term “GDM-like fuels” in the text, although it is understood what it refers to, it would be appropriate to indicate it in the text.

You could indicate from which article you have drawn this conclusion, indicated in the introduction (page 2, line 63-63): “When the balance between FA availability and mitochondrial function is tipped, lipid accumulates in droplets and organelles”. In the indicated ones (citations 19 and 20, not found)

Page 15, line 492-493, I do not understand well what you mean in this sentence, I do not know if the idea is poorly expressed: “By design, the study only includes consenting females, but both male and female placentas were used for CTB isolation”

Page 16: Has the Primary Cytotrophoblast isolation method been contrasted in other publications? If so, I think it would be appropriate to include some bibliographic citation and if not, some indicator of viability of these cells should be introduced. Page 3, line 113: Clarify the terms doubling time, or fold change

Round 2

Reviewer 3 Report

Comments and Suggestions for Authors

the manuscript can be accepted in the current form